# Grazing and Cultivated Grasslands Cause Different Spatial Redistributions of Soil Particles

**DOI:** 10.3390/ijerph16152639

**Published:** 2019-07-24

**Authors:** Jinsheng Li, Jianying Shang, Ding Huang, Shiming Tang, Tianci Zhao, Xiaomeng Yang, Qian Zhang, Kesi Liu, Xinqing Shao

**Affiliations:** 1Department of Grassland Science, China Agricultural University, Beijing 100193, China; 2Department of Water and Soil Science, China Agricultural University, Beijing 100193, China; 3National Field Station of Grassland Ecosystem in Guyuan, Zhangjiakou 075000, China

**Keywords:** grazing grassland, cultivated grassland, soil particle, sustainability use, desertification

## Abstract

The distribution of soil particle sizes is closely related to soil health condition. In this study, grasslands under different grazing intensities and different cultivation ages grasslands were selected to evaluate the dynamics of soil particle size redistribution in different soil layers. When the grazing intensity increased, the percentage of 2000~150-μm soil particles in the 0–10-cm soil layer decreased; 150~53-μm soil particles remained relatively stable among the grazing intensities—approximately 28.52%~35.39%. However, the percentage of less than 53-μm soil particles increased. In cultivated grasslands, the larger sizes (>53 μm) of soil particles increased and the smaller sizes (<53 μm) decreased significantly (*p* < 0.05) in the 0–10 cm-soil layer with increasing cultivation ages. The increase in small soil particles (<53 μm) in topsoil associated with grazing intensity increased the potential risk of further degradation by wind erosion. The increase in big soil particles (>53 μm) in topsoil associated with cultivation ages decreased the soil capacity of holding water and nutrient. Therefore, to maintain the sustainability of grassland uses, grazing grasslands need to avoid heavy grazing, and cultivated grasslands need to change current cultivation practices.

## 1. Introduction

Grasslands are the largest terrestrial landform in China, accounting for nearly 40% of the total land area [1]. Currently, the principal methods of grassland utilization include keeping grasslands as grazing lands or mowing lands and converting grasslands into croplands. During the past half century, approximately 8 × 10^6^ ha of native grasslands in China have been converted to croplands [2]. These different grassland uses affect soil nutrients, soil structure, and the distribution and stability of soil aggregates [3]. Numerous studies have reported that appropriate land use methods of can improve soil structure, increase soil permeability, optimize soil hydrothermal conditions and enhance soil resistance to external environmental changes; however, other methods of land utilization accelerate the loss of soil nutrients and lead to soil degradation and even desertification [4,5,6]. The fates of different grassland utilization methods are closely linked to the redistribution of soil particles, which regulates the states of the soil water, gas and heat [7,8]; affects the activity and type of soil microbes and other soil fauna [9,10]; and determines the nutrient levels of the soil [11]. Some studies have used the distribution of soil particles to evaluate the dynamics of soil texture, erosion resistance and soil fertility at study sites [12,13]. Different land use patterns and years affect the redistribution of soil particles in different soil layers [14,15,16,17]. However, information considering the effects of land use on the spatial redistribution of soil particles is still rare. Grasslands used for different purposes are very common in the Bashang Plateau region, an agro-pastoral ecotone in northern China. The grasslands of this region have been grazed or converted into croplands for a long time [18]. Changes in the intensity of grazing or the sustained cultivation of grasslands could affect the redistribution of different soil particles, which might promote or degrade the soil health. Scientific utilization and management of different grassland uses in the agro-pastoral ecotone requires a better understanding of soil health condition through evaluating the distribution of soil particles. Therefore, the objective of this study was to examine the effects of the different land uses on the spatial redistribution of soil particles.

## 2. Materials and Methods

### 2.1. Experimental Sites

The study was conducted in the National Field Station of the Grassland Ecosystem in Guyuan County, Hebei Province, China (41°44′ N, 115°42′ E), which is located in the agro-pastoral ecotone. The region has a typical semiarid continental monsoon climate with an annual mean precipitation of 400 mm, approximately 80% of which falls in the growing season from July to September. The annual evaporation capacity is 1700–2300 mm. The mean annual temperature ranges from 2 to 8 °C. The dominant plant in grasslands is Chinese wildrye (*Leymus chinensis Tzvel*), accompanying rare species Krylov Needlegrass (*Stipa krylovii*) and *Potentilla anserina*. Over the past several decades, some grasslands in this region have been converted to croplands of naked oat (*Avena nuda*) and flax (*Linum usitatissimum* L) in a rotation scheme. The soil in this region is chestnut soil or Calciustepts by Chinese Soil Taxonomy and US Soil Taxonomy, respectively.

### 2.2. Experimental Design and Soil Sampling

The three main grassland use patterns in this agro-pastoral ecotone are enclosure grasslands, grazing grasslands, and cultivated grasslands. The original use of these lands was all grazing grasslands. In this study, six grassland uses were selected, including three grazing grasslands with different intensities (10-year enclosure grasslands without grazing, G10); moderate grazing grasslands with 10 goats ha^−1^, MG; heavy grazing grasslands with 14 goats ha^−1^, HG) and three cultivated grasslands with different cultivation ages (5 years, C5; 10 years, C10; 15 years, C15). Fifty-hectare grasslands were fenced to prevent disturbances from grazing animals in July 2007. The initial grassland conditions were 25-cm average herbage height and 50–70% vegetation coverage. Grazing had been excluded for 10 years before the sampling year of 2016. In 2012, in order to evaluate the effects of grazing intensity on grasslands, approximately 4 ha of enclosure grasslands were assigned for moderate (MG) and heavy grazing (HG) experiments, and the area of each grazing intensity was 2 ha. The initial conditions of the assigned grasslands were approximately 45 cm in average height and over 85% vegetation coverage in July. The grazing method was continuous grazing, and the MG and HG treatments both included two, one-hectare grazing paddocks. The grazing period was from July to September. Forty-eight, two-year-old, healthy, local goats were assigned to the two treatments: 20 goats with an average initial weight of 45.7 ± 0.5 kg for MG (10 goats per paddock) and 28 goats with an average initial weight of 45.3 ± 0.7 kg for HG (14 goats per paddock). Grazing events were initiated on 6 July of 2012, 2013, 2014, 2015 and 2016. All goats remained in the paddocks during the grazing period.

The cultivated grasslands were continuously rotationally cultivated for 5 years, 10 years, and 15 years before they were sampled in 2016. Each cultivation age of grassland included two similar paddocks, and the area of each paddock was approximately 2–3 hectares. The rotation crops were *Avena chinensis* and *Linum usitatissimum* L. *Avena chinensis* was the cultivated crop in 2016. Conventional tillage was performed in the cultivated grasslands using disk at a depth of 20 cm and rotary tiller (Worksaver, Inc., Litchfield, IL, USA) at a depth of approximately 40 cm one day prior to planting the crop. The crop was planted and harvested at the end of April and August, respectively. The enclosure grasslands and grazing grasslands did not receive fertilization, and the cultivated grasslands received 50 kg ha^−1^ year^−1^ diammonium phosphate (N: 9 kg ha^−1^ year^−1^; P: 23 kg h^−1^ year^−1^) at the early growth stage of the crop. All study sites exhibited the same topography, soil series, and climate conditions without irrigation.

Soil samples were collected in July 2016. Three 5 m × 5 m quadrats were randomly selected from each experimental field, and 10 sample points within each quadrat were randomly selected to collect soil core soil using a handheld 3.5-cm-diameter soil auger at 0–10 cm, 10–20 cm, 20–40 cm, 40–60 cm and 0–60 cm depths. Samples from the same soil layer within the same sampling quadrat were combined into one composite sample and then separated into three soil subsamples as replicates. Total nine subsamples for each soil layer were collected from each experimental field. These samples were hand sorted to remove rocks and visible plant materials, dried at 65 °C to constant weight, and then passed through a 2-mm screen for particle size analysis.

### 2.3. Particle Size Analysis

A 0.4 g soil sample was weighed. Then, enough 30% hydrogen peroxide (H_2_O_2_) was added to remove the organic matter from the sample, and a sufficient amount of hydrochloric acid (HCl) was added to remove the carbonate content. Ultra-pure water was then added to dilute the treated sample, and the supernatant was disposed to remove the acid from the solution. The same process was repeated until the pH of supernatant was 6.5~7.0. Then, the nearly neutral soil sample was added to dispersant hexametaphosphate (Na_2_PO_3_)_6_ and boiled for 1 h in an electric sand bath. After that process, the volume percentages of different primary soil particle sizes in the soil samples were measured using a Mastersizer 2000 laser particle analyzer (Malvern, UK). The soil particle sizes were divided into 6 groups of 2000~250 μm, 250~150 μm, 150~53 μm, 53~20 μm, 20~2 μm and less than 2 μm.

### 2.4. Statistical Analysis

The effects of different grassland use on the spatial redistribution of soil particle sizes were assessed using the PROC MIXED procedure of SAS (SAS Inst. Inc. 2001, Cary, NC, USA). Grazing grasslands and cultivated grasslands were analyzed using a separate ANOVA. Grazing intensity or cultivation age was considered fixed effects. Replication was considered a random effect. The PDIFF test of the LSMEANS procedure was used to compare the means. All data reported are least squares means. Treatments were considered significantly different when *p* ≤ 0.05.

## 3. Results

### 3.1. Effects of Grazing Intensity on the Redistribution of Soil Particle Sizes

The distribution percentage of soil particle sizes in the 0–60-cm soil layer under different grazing intensities was significantly different (Figure 1). When the grassland type changed from 10-year enclosure grassland (G10) to heavy grazing grassland (HG), the quantity of larger (>53 μm) soil particles decreased, and the quantity of smaller (<53 μm) particles increased significantly (*p* < 0.05). In the G10 and moderate grazing grasslands (MG), the percentages of 150~53-μm soil particles were the greatest at 37.39% and 31.31%, respectively. However, the percentage of 20~2-μm soil particles was the maximum among the six groups in HG, up to 32.42%.

When 0–60-cm soil depth was divided into four different layers (0–10, 10–20, 20–40 and 40–60 cm), the soil particle sizes showed varied dynamics of spatial distribution under the different grazing intensities (Table 1). The percentage of 2000~250-μm soil particles in the 0–10=cm soil layer decreased from 17.12% in G10 to 5.56% of HG. In the 10–60-cm soil layer, the percentage of 2000~250-μm soil particles underwent no significant change between MG and G10 but underwent a significant decrease from MG to HG (*p* < 0.05).

The percentage of 250–150-μm soil particles in the 0–20 cm soil layers was significantly higher in G10 than those in MG and HG. In the 20–60-cm soil layers, the percentages of 250–150 μm soil particles in G10 and MG were not significantly different but were both significantly higher than the percentage in HG. The percentage of 150~53-μm soil particles remained relatively stable among the grazing intensities, approximately 28.52%~35.39% (Table 1).

However, the percentage of 53~20-μm and 20–2-μm soil particles increased with increasing grazing intensities from G10 to HG. In the 0–10-cm soil layer, the percentage of 20~2-μm soil particles nearly doubled, from 16.78% to 31.29%, when grasslands changed from G10 to HG. In the 20–40 cm and 40–60-cm soil layers, the percentage of 20–2-μm soil particles had no significant differences between G10 and MG but was significantly lower than that in HG.

For less than 2–μm soil particles, the effects of grazing intensity mainly occurred in the 0–10-cm soil layer, and HG significantly increased the percentage of this size of soil particle in this layer. The percentage of less than 2–μm soil particles in the 10–60-cm soil layers was not different among the different grazing intensities.

Within the same grazing intensity, the percentage of soil particles showed some fluctuation among some different layers but they were not as strong as among grazing intensities (Table 1). In G10, the percentage of 2000~250-μm soil particles in the 0–10-cm soil layer was greater than in the 10–60-cm soil layer, but MG resulted in a lower percentage of this particle size in the 0–10-cm soil layer compared to that in the 10–60-cm soil layers. In HG, the percentage of 2000~250-μm soil particles in the 0–40 cm layer was significantly greater than that in the 40–60-cm layer. Within G10 or HG, the percentage of 250~150-μm soil particles among the different soil layers was not significantly different, but in MG, the percentage of 250~150-μm soil particles in the 0–20 cm soil layer was lower than that in the 20–60-cm soil layer. As for the percentage of 150~20-μm soil particles, any grazing intensity did not show significant difference among different soil layers. The percentage of 20~2-μm soil particles remained relatively stable among different soil layers in G10 and HG but was greater in the 0–20 cm soil layers than in the 20–60-cm soil layers in MG. There was no difference among soil layers in any grazing intensity.

### 3.2. Effects of Cultivation Age on the Redistribution of Soil Particle Sizes

After the grasslands had been converted into cropland, the larger sizes (>53 μm) of soil particles increased, and the smaller sizes (<53 μm) decreased significantly (*p* < 0.05) in the 0–60-cm soil layer with increasing cultivation age (Figure 2). In the 5-year cultivated grassland (C5), the percentage of 20~2-μm soil particles was the greatest, 33.15%, but the percentage of this particle size decreased to 22.58% and 17.18%, when the cultivation age increased to 10 years (C10) and 15 years (C15), respectively. The percentage of 150~53-μm soil particles reached a maximum in C10 and C15, up to 32.42% and 37.01%, respectively. The percentage of less than 2-μm soil particles did not significantly change after the grasslands had been cultivated for 10 years, approximately 2.79–3.11%.

When the 0–60-cm soil layer was divided into four different layers, the soil particle sizes showed varied fluctuations in the soil layers under different cultivation ages (Table 2). The percentage of 2000~250-μm soil particles in different soil layers increased significantly as the cultivation age of the grasslands increased from 5 years to 15 years. In the 0–10-cm and 10–20-cm soil layers, the percentages of 2000–250 μm soil particles in C10 and C15 were not significantly different but were both significantly higher than that in C5. In the 40–60-cm soil layer, the percentage of 2000–250-μm soil particles in C15 was twice that in C5, 23.52% vs. 11.49% (Table 2). The percentage of 2000~250-μm soil particles was not significantly different among the different soil layers in C5 or C10. However, this percentage in C15 was smaller in the 0–10-cm and 10–20-cm soil layers than in the 20–60-cm soil layers.

The percentage of 250~150-μm soil particles in the different soil layers also increased significantly as the cultivation age increased (Table 2). In the 0–10-cm soil layer, the percentages of 250–150 μm soil particles in C10 and C15 were 81% and 134% higher than those in C5, respectively. In the 20–60-cm soil layers, the 250~150-μm soil particles were not significantly different between C10 and C15, but they were both significantly higher than the level in C5. In a given cultivation age, the percentage of 250~150-μm soil particles was not significantly different among the different soil layers.

The content of 150~53-μm soil particles in the 0–10-cm soil layer was not significantly different in C5 and C10 but was increased in C15. In the 20–60-cm soil layers, the content of 150–53-μm soil particles was not different in C10 and C15, but these contents were both higher than that in C5. However, the percentage of soil particles smaller than 53 μm generally decreased with increasing cultivation age (Table 2). In the 0–10-cm and 20–40 cm soil layers, the percentage of 53–20-μm soil particles decreased in an orderly manner from C5 to C10 to C15. In the 10–20-cm and 40–60-cm soil layers, the percentage of 53–20-μm soil particles was not different between C10 and C15, but these percentages were both lower than that in C5 (Table 2).

The percentage of 20~2-μm soil particles decreased from C5 to C10 to C15 in every soil layer. The percentage of less than 2 μm soil particles in a given soil layer was not different between C10 and C15, but these percentages were lower than that in C5. In C5, the percentage of less than 2-μm soil particles in the 0–10-cm soil layer was lower than those in the other 10–60-cm soil layers. However, the percentage of less than 2 μm soil particles was not different among the different soil layers in C10 or C15.

## 4. Discussion

### 4.1. Effects of Grazing Intensity on the Redistribution of Soil Particle Sizes

Grazing mainly affects the soil properties of the grasslands in three main ways: foraging, trampling and fouling [19]. Different grazing intensities resulted in different changing trends of soil particle size. Some studies have reported that the contents of coarse and medium-grain sand increased and the content of silt and clay decreased in long-term heavy-grazing grasslands [20,21,22]; it was theorized that overgrazing decreases coverage and accelerates wind erosion. Fu et al. [23] found that increasing grazing intensity in desert grasslands easily leads to soil exposure that then causes wind erosion to decrease the amount of small soil particles and relatively increase the proportion of larger soil particles on the surface. In some grasslands with relatively high precipitation and a small influence of wind erosion, the change in the soil texture component in grazing grasslands is probably due to trampling and other factors. Lin et al. [24] found in a simulated experiment of trampling and precipitation that high-intensity trampling can slow the coarsening of topsoil with a certain amount of precipitation (>192 mm), and under abundant water conditions (266 mm), the topsoil has a higher ratio of clay to sand. The degree of animal trampling on soil increases with increasing grazing intensities, and this trampling intensity increase may be one of the main reasons for the increase in small soil particle sizes. Yang et al. [25] found that the trampling of cattle and sheep can accelerate the redistribution of soil particles and increase the clay content on the soil surface. With increasing clay contents, the nutrient preservation capacity can be improved, as was proven by Zhang et al. [26]. Our results verified these findings to some degree. In our study site, the annual rainfall reached 400 mm. Our results show that the percentage of sand (greater than 53 µm soil particles) decreased and the percentage of silt (53-2 µm soil particles) and clay (soil particles smaller than 2 μm) increased when the grazing intensity increased. Hewins et al. [27] also reported that most of soil organic carbon was bond with soil particles smaller than 53 μm and the content of soil particles smaller than 53 μm increased in the top soil under grazing disturbance. The small soil particle sizes of the topsoil caused by heavy grazing were easily eroded by wind. Yan et al. [2] reported that wind erosion is a key process that causes soil degradation in the semiarid steppe regions of northern China and found that dry aggregate fractions of <0.2 mm were selectively depleted by wind erosion and exhibited an exponential decrease in the residual soils with increasing wind erosion intensity. If this grazing intensity is maintained, the desertification of grasslands would occur mostly due to the removal of clay, with sand remaining after wind erosion.

### 4.2. Effects of Cultivation Age on the Redistribution of Soil Particle Sizes

The distribution of soil particles is an important soil physical characteristic that affects soil hydraulic characteristics, soil fertility and soil erosion [28]. Changes in land use change the redistribution of different soil particle sizes to some extent [29], and different vegetation types also have a significant effect on the particle size distribution of different soil layers [30]. Our results showed that the percentage of greater than 53 µm soil particles significantly increased and the percentage of less than 53 µm soil particles significantly decreased with increasing cultivation age. This trend might be caused mainly by wind erosion and human tillage activity, which cause soil fragmentation. Six et al. [31] reported that the cultivation process generally broke macroaggregates (>250 µm) into microaggregates (<250 µm). These microaggregates on the soil surface were easily reduced by wind erosion or rainfall [32]. In the agro-pastoral ecotone of China, Hu et al. [33] found that wind erosion is usually higher in cultivated lands than in grasslands. In northern Europe, Sheehy et al. [34] also reported that the effect of wind erosion on traditional tillage systems is higher than that on no-tillage systems. The effects of wind erosion accumulated with increasing cultivation age. The more than the <53 μm soil particles are lost, the more that the soil particles on the surface coarsen. The physical disturbances and fragmentation of soil by human cultivation, as well as the decrease in aboveground biomass after cultivation, further aggravated the degree of soil wind erosion [35]. In our study, we found that the percentage of soil particle sizes in a given soil layer five years after cultivation (C5) was significantly different from C10 and C15 and the percentage of soil particle sizes had no significant difference in any soil layer between C10 and C15. This indicated that the obvious reduction of aboveground biomass made the surface soil lack cover protection after grassland conversion to cultivated land. The enhanced soil wind erosion significantly decreased the percentage of soil particles smaller than 53 μm and resulted in a significant increase in the percentage of 2000-53 μm soil particles in the first few years of grassland cultivated. After 10 to 15 years of cultivated, the soil particle sizes stabilized; therefore, there was no significant change in the percentage of soil particles among the cultivation ages.

## 5. Conclusions

Grassland uses affected the redistribution of soil particle sizes. For grazing grasslands, the percentage of greater than 53-µm soil particles within the 0–60-cm grassland soil depth decreased significantly with increasing grazing intensity, but the content of soil particles smaller than 53 μm increased significantly. Considering the different soil layers in the 0–60-cm soil depth, the effect of grazing intensity on the redistribution of soil particle sizes mainly occurred in the 0–10-cm layer. The increase in small soil particles in topsoil with grazing intensity, especially under heavy grazing, increased the potential risk of further degradation by wind erosion. After the cultivation of grasslands, the content of greater than 53 µm soil particles in the 0–60-cm soil depth significantly increased, and the contents of less than 53 µm soil particles significantly decreased with increasing cultivation age. The fluctuation of different soil particle sizes with the change of cultivation age mostly occurred in the 0–10-cm and 10–20-cm soil layers. The increase in large soil particles with the cultivation age indicated that the soil gradually tends towards desertification and the soil health is deteriorating. Therefore, to ensure the sustainability of grassland uses, grazed grasslands need to avoid heavy grazing, and cultivated grasslands need to change current farming practices.

## Figures and Tables

**Figure 1 ijerph-16-02639-f001:**
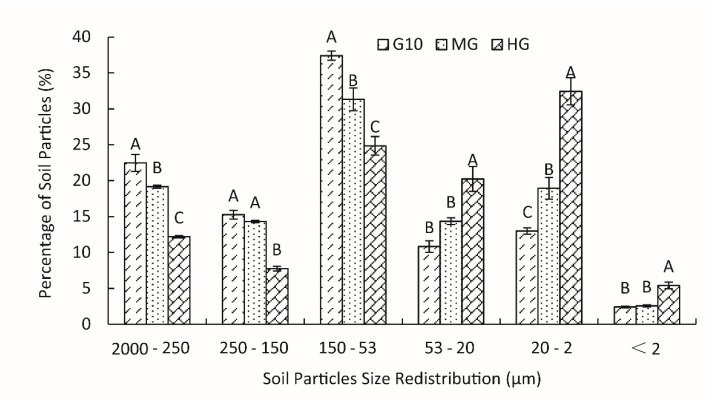
The redistribution of soil particles under different grazing intensities. Different uppercase letters indicate significant differences among the different grazing intensities at *p* ≤ 0.05. G10 represents 10-year enclosure grassland without grazing; MG represents moderate grazing grassland with 10 goats ha^−1^; HG represents heavy grazing grassland with 14 goats ha^−1^.

**Figure 2 ijerph-16-02639-f002:**
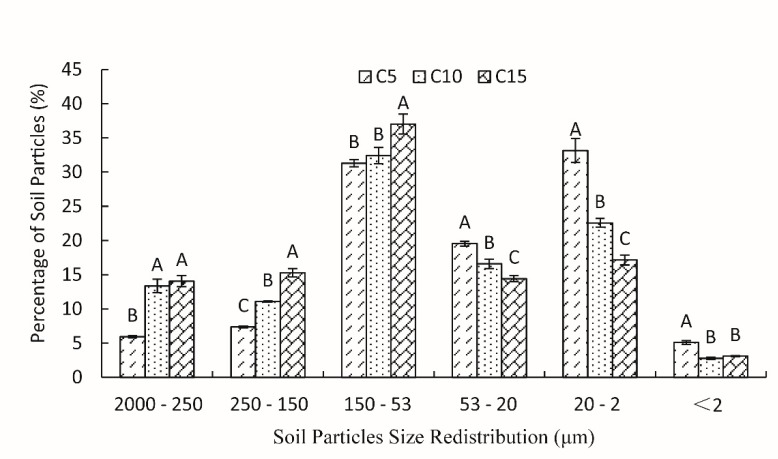
The redistribution of soil particles under different cultivation ages. Different uppercase letters indicate significant differences among the different cultivation ages at *p* ≤ 0.05. C5 represents the cultivated grasslands with five cultivated years; C10 represents the cultivated grasslands with 10 cultivated years; C15 represents the cultivated grasslands with 15 cultivated years.

**Table 1 ijerph-16-02639-t001:** The redistribution of soil particle sizes within soil layers under different grazing intensities.

Grazing Intensity	Soil Depth			
	0~10 cm	10~20 cm	20~40 cm	40~60 cm
	2000~250 μm			
G10	17.12 aA *	13.89 aB	13.25 aB	12.66 aB
MG	11.68 bB	12.99 aAB	14.61 aA	14.14 aA
HG	5.56 cAB	7.80 bA	6.36 bA	3.32 bB
	250~150 μm			
G10	15.83 aA	14.00 aA	13.30 aA	15.41 aA
MG	10.28 bB	9.15 bB	11.30 abAB	13.64 aA
HG	7.59 bA	7.87 bA	6.87 bA	4.54 bA
	150~53 μm			
G10	34.73 aA	35.39 aA	35.17 aA	36.47 aA
MG	33.62 aA	28.80 bB	32.64 abA	34.60 aA
HG	30.26 aA	27.43 bA	29.14 bA	28.52 bA
	53~20 μm			
G10	13.09 bA	14.17 bA	15.22 aA	11.44 bA
MG	19.94 aA	19.77 aA	15.75 aB	12.57 bB
HG	20.97 aAB	18.22 abAB	17.66 aB	21.41 aA
	20~2 μm			
G10	16.78 cA	19.65 bA	19.74 bA	20.39 bA
MG	22.27 bB	26.05 aA	22.23 bB	21.83 bB
HG	31.29 aA	34.84 aA	33.67 aA	37.26 aA
	<2 μm			
G10	2.44 bA	2.91 aA	3.32 bA	3.63 aA
MG	2.22 bA	3.24 aA	3.47 bA	3.22 aA
HG	4.32 aA	3.86 aA	6.29 aA	4.93 aA

* For a given particle size, different lowercase letters within the same column indicate significant differences among different treatments at *p* ≤ 0.05; different uppercase letters in the same row indicate significant differences among different soil layers at *p* ≤ 0.05.

**Table 2 ijerph-16-02639-t002:** The redistribution of soil particle sizes within soil layers under different cultivation ages.

Cultivation Age	Soil Depth			
	0~10 cm	10~20 cm	20~40 cm	40~60 cm
	2000~250 μm			
C5	11.42 bA *	8.51 bA	12.00 bA	11.49 bA
C10	17.65 aA	20.15 aA	18.90 abA	18.82 abA
C15	15.25 abC	17.84 aBC	26.77 aA	23.52 aAB
	250~150 μm			
C5	7.25 cAB	5.15 bB	7.10 bAB	8.43 bA
C10	13.14 bA	14.54 aA	14.51 aA	13.64 aA
C15	16.96 aA	16.19 aAB	14.52 aAB	13.38 aB
	150~53 μm			
C5	28.77 bA	22.58 cB	22.43 bB	25.62 bAB
C10	31.44 bA	30.36 bA	30.99 aA	32.45 abA
C15	39.78 aA	38.74 aA	35.66 aA	35.40 aA
	53~20 μm			
C5	21.44 aAB	23.13 aA	18.66 aB	17.70 aB
C10	16.10 bA	14.23 bAB	14.27 bAB	12.83 bB
C15	11.99 cA	11.19 bAB	9.33 cC	10.78 bB
	20~2 μm			
C5	26.86 aB	34.78 aA	33.60 aAB	31.43 aAB
C10	19.06 bA	18.25 bA	18.92 bA	19.50 bA
C15	13.65 cA	13.52 cAB	11.70 cB	14.27 bA
	<2 μm			
C5	4.26 aB	5.85 aA	6.20 aA	5.33 aA
C10	2.60 bA	2.47 bA	2.41 bA	2.77 bA
C15	2.38 bA	2.53 bA	2.03 bA	2.65 bA

* For a given particle size, different lowercase letters within the same column indicate significant differences among different treatments at *p* ≤ 0.05; different uppercase letters in the same row indicate significant differences among the different soil layers at *p* ≤ 0.05.

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
