# Peer review of "Grazing and Cultivated Grasslands Cause Different Spatial Redistributions of Soil Particles"

_ijerph, 2019, doi:10.3390/ijerph16152639_

Round 1

Reviewer 1 Report

I had a problem reading through the text until the conclusions to understand what you meant by "reclaimed grasslands". I wasn't sure if you meant grassland converted to crops, or crops converted to grasslands. I think this should be clearer.

My other main suggestion for improvement is to try and have some of the results currently in the tables as graphs. I found it incredibly difficult to follow and visualise what was going on.

A final suggestion would be to incorporate more references to western studies. There have been many in Europe and I believe in the US, and some more could be mentioned and compared.

Further comments in the PDF attached.

Reviewer 2 Report

The manuscript “Grazing and reclamation grasslands cause different spatial redistributions of soil particles” is a well-written research article about the effects of different management regimes on the soil particle size distribution in permanent grassland. The well-presented and well-discussed results help better understand how soil degradation could be reduced. Nowadays, this aspect of soil protection becomes more and more relevant within a changing world of increasing land use conflicts, because it reduces the losses of utilizable soil in the long-term. Therefore, I would recommend to accept this article after it has been adequately revised according to a few major and minor comments from my side.  

L19 and whole manuscript: please consider writing “ha-1” instead of “/ha”, because it is the common scientific form.

L22 Please write “250~150 μm” instead of “250~150um”.

L23 Please write “20~2 μm” instead of “20-2um”. And please write “2 μm” instead of “2um”.

Figures 1 and 2: I would recommend writing the unit in the axis title - for example: “Soil particles size redistribution (μm)” instead of writing the unit “μm” after each class of soil particles at the axis (for example: “2000 – 250” instead of “2000-250μm”. If the authors do not agree, I would recommend to write empty spaces between the numbers and the values.

L25 Please consider writing “P < 0.05” instead of “P<0.05”.

L46-47 The sentence “In a simulation experiment, Zhang 46 et al. [14] found that the ability of

soil to preserve nutrients increased with increasing clay contents.” confuses me. This is because the clay content is not mentioned in the sentences before (“The fates of different grassland utilization methods are closely linked to the redistribution of soil particles (…)”) and after it (“Different land use patterns and years affect the redistribution of soil particles in different soil layers [15-18].”) both of which are about soil particles in particular – and not explicitly about clay content. What about moving the sentence on clay content/nutrient preservation from L46 to L238 after the sentence “Yang et al. [26] found that the trampling of cattle and sheep can accelerate the redistribution of soil particles and increase the clay content on the soil surface.”? I suppose it would perfectly fit there, because then it could be argued that, together with the clay content, also the nutrient preservation capacity can be improved, as was proven by Zhang et al. (…).   

L88 The sentence “A roofed building and water tank were built near the edge of the inside paddock.” could, in my opinion, be omitted. I consider this comment as optional.

L95 “rotary tiller” instead of “Worksaver rotary tiller” (redundant, because the company’s name is provided in brackets afterward).

L98 “50 kg/ha/year diammonium phosphate” please also provide exact amounts of N and P, in brackets would be ok, in my opinion.

L99 Please provide the basic annual average temperatures and annual precipitation values for the observation period, if available (which I assume should be available, because the authors state that “All study sites exhibited the same (…) climate conditions (…)”.

L116 “Mastersizer 2000 laser particle analyzer.” Please provide the company’s name and country!

L121-122 Please consider also writing the model, not only describing it. And please describe the model a bit further, for example: what about the intercept? And what about interactions between grazing intensity and reclamation age? Did you measure interactions? If not, why not?

L278 “53 μm” instead of “53μm”

L279 and throughout the manuscript: I would suggest writing “soil particles smaller than 53 μm” instead of “less than 53 μm soil particles”.

References:

I would recommend a brief discussion (maybe 2 to 5 sentences, not more) of the results of this study in context of the following two references: Zhang et al. (2019) (DOI: 10.1016/j.scitotenv.2018.08.211) and Hewins et al. (2018) (DOI: 10.1038/s41598-018-19785-1). Both of these studies provide up-to-date information on the research field of soil particle size distribution in different grassland regimes.

Round 2

Reviewer 1 Report

I still have a problem with the use of the term "reclamation grasslands". According to the Oxford Dictionary, "Reclamation is the process of changing land that is unsuitable for farming (or building) into land that can be used". Grazing is farming too, so this term is not appropriate here.

As said in my first review, I find the use of "reclamation grasslands" confusing, and having checked with my Grasslands Specialist (and English native speaker) colleague, we agree that calling them "Cultivated grasslands" would be much clearer for readers around the World.

Otherwise, just minor changes suggested in the document attached.

Author Response

Response to Reviewer 1 Comments

Point 1: I still have a problem with the use of the term "reclamation grasslands". According to the Oxford Dictionary, "Reclamation is the process of changing land that is unsuitable for farming (or building) into land that can be used". Grazing is farming too, so this term is not appropriate here.

As said in my first review, I find the use of "reclamation grasslands" confusing, and having checked with my Grasslands Specialist (and English native speaker) colleague, we agree that calling them "Cultivated grasslands" would be much clearer for readers around the World.

Response 1: Thanks for your suggestion. We accepted your suggestion. Revised, and please see the new version.

Point 2: Oxford Dictionary - "Reclamation: the process of changing land that is unsuitable for farming (or building) into land that can be used". Grazing is farming too, so this term is not appropriate here.

As said in my first review, I find the use of "reclamation grasslands" confusing, and having checked with my Grasslands Specialist (and English native speaker) colleague, we agree that calling them "Cultivated grasslands" would be much clearer for readers.

Response 2: Thanks for your suggestion. Revised, and please see the new version.

Point 3: I think this would be clearer:

In this study, grasslands under different grazing intensities and different cultivation ages grasslands were selected...

Response 3: Thanks. Revised, and please see Line 15-17.

Point 4: cultivated

Response 4: Thanks. Revised, and please see line 21.

Point 5: cultivation

Response 5: Thanks for your suggestion. Revised, and please see line 25.

Point 6: cultivation

Response 6: Thanks. Revised, and please see line 27.

Point 7: replace by “the effects of land use”

Response 7: Thanks. Revised, and please see line 47.

Point 8: delete “utilization” “grasslands” “reclamation” “grass”

Response 8: Thanks. Revised, and please see line 50-54.

Point 9: r (grazing)

Response 9: Thanks for your suggestion. Revised, and please see line 50.

Point 10: Stipa krylovii

Response 10: Thanks. Revised, and please see line 64.

Point 11: anserina

Response 11: Thanks. Revised, and please see line 64.

Point 12: of

Response 12: Thanks. Revised, and please see line 65.

Point 13: cultivated

Response 13: Thanks. Revised, and please see line 70.

Point 14: delete “Reclamation grasslands referred to grasslands which were converted into croplands.”

Response 14: Thanks. Revised.

Point 15: delete “patterns”

Response 15: Thanks. Revised, and please see line 70.

Point 16: was

Response 16: Thanks. Revised, and please see line 70.

Point 17: cultivated

Response 17: Revised, and please see line 74.

Point 18: In line 83 it says "continuous grazing", but it seems it started in July each year. For how long?

Response 18: We started the grazing events after the height of grassland was higher than 1000px. The grazing period was from July to September. We mentioned it at line 83.

Point 19: cultivated

Response 19: Thanks. Revised, and please see line 88.

Point 20: cultivation

Response 20: Thanks. Revised, and please see line 89.

Point 21: cultivated

Response 21: Thanks. Revised, and please see line 92.

Point 22: elsewhere as kg ha-1 yr-1

Response 22: Thanks. Revised, and please see line 96.

Point 23: again kg ha-1 yr-1

Response 23: Thanks. Revised, and please see line 96.

Point 24: cultivated

Response 24: Thanks. Revised, and please see line 121.

Point 25: cultivation

Response 25: Thanks. Revised, and please see line 121.

Point 26: Consider braking the paragraph after each particle size to make it easier to read?

Response 26: Thanks for your suggestion. Revised, and please see Line 140-159

Point 27: T

Response 27: Thanks. Revised, and please see line 175.

Point 28: cultivation

Response 28: Thanks. Revised, and please see line 177.

Point 29: Again, consider breaking the paragraph for each particle size?

Response 29: Thanks for your suggestion. Revised, and please see Line 193-223

Point 30: delete “in an orderly manner”

Response 30: Thanks for your suggestion. Revised, and please see Line 218.

Point 31: repeated!

Response 31:  Thanks for your finding. Deleted, and please see line 244.

Point 32: cultivation

Response 32: Thanks. Revised, and please see line 259.

Point 33: cultivation

Response 33: Thanks. Revised, and please see line 265.

Point 34: cultivation

Response 34: Thanks. Revised, and please see line 277.

Point 35: cultivation

Response 35: Thanks. Revised, and please see line 292.

Point 36: cultivation

Response 36: Thanks. Revised, and please see line 297.

Point 37: grazed grasslands

Response 37: Thanks. Revised, and please see line 298.

Point 38: cultivated

Response 38: Thanks. Revised, and please see line 299.

Point 39: what are these "previous farming practices"? Do you mean back to grazing or to a non-tillage seeding?

Response 39: Thanks for reminding. We changed it to “current farming practices”, and please see line 299.

Reviewer 2 Report

Dear authors,

thank you very much for thoroughly addressing all of my comments!

Kind regards!

Author Response

Thank you for your reply